# Radial alignment of microtubules through tubulin polymerization in an evaporating droplet

Jakia Jannat Keya[1], Hiroki Kudoh[2], Arif Md. Rashedul Kabir[1], Daisuke Inoue[3], Nobuyoshi Miyamoto[4], Tomomi Tani[5], Akira Kakugo[1,6]*, Kazuhiro Shikinaka[7]*

**1** Faculty of Science, Hokkaido University, Sapporo, Hokkaido, Japan, **2** Graduate School of Engineering, Tokyo University of Agriculture and Technology, Koganei, Tokyo, Japan, **3** Department of Human Science Faculty of Design, Kyushu University, Fukuoka, Japan, **4** Department of Life, Environment and Materials Science, Fukuoka Institute of Technology, Fukuoka, Japan, **5** Eugene Bell Center for Regenerative Biology and Tissue Engineering, Marine Biological Laboratory, Woods Hole, MA, United States of America, **6** Graduate School of Chemical Sciences and Engineering, Hokkaido University, Sapporo, Hokkaido, Japan, **7** Research Institute for Chemical Process Technology, National Institute of Advanced Industrial Science and Technology, Sendai, Miyagi, Japan

* kakugo@sci.hokudai.ac.jp (AK); kaz.shikinaka@aist.go.jp (KS)

**Data Availability Statement:** All relevant data are within the paper and its Supporting Information files.

## Abstract

We report the formation of spherulites from droplets of highly concentrated tubulin solution via nucleation and subsequent polymerization to microtubules (MTs) under water evaporation by heating. Radial alignment of MTs in the spherulites was confirmed by the optical properties of the spherulites observed using polarized optical microscopy and fluorescence microscopy. Temperature and concentration of tubulins were found as important parameters to control the spherulite pattern formation of MTs where evaporation plays a significant role. The alignment of MTs was regulated reversibly by temperature induced polymerization and depolymerization of tubulins. The formation of the MTs patterns was also confirmed at the molecular level from the small angle X-ray measurements. This work provides a simple method for obtaining radially aligned arrays of MTs.

## Introduction

Self-assembly is the preferred means of nature to create structures at various scales from discrete components [1]. The self-assembly processes result in fascinating patterns or ordered structures that are spatially and functionally distinct with the ability to exhibit emergent functions [2]. In living systems, self-assembly occurs under non-equilibrium condition leading to patterns or assembled structures that play indispensable roles in ensuring cellular events required for development and survival [3]. One of the prominent examples of self-assembly in nature is the cytoskeletal network of filamentous proteins, such as actin filament and MT. MT is a cylindrical protein assembly with a diameter of approximately 25 nm formed by the polymerization of globular tubulin dimers [4]. Assembly of the MT filaments produces a variety of structures in the cell that play vital roles in cell division, intracellular transport, cell shape

**Funding:** Fund receiver: Akira Kakugo Grant-in-Aid for Scientific Research on Innovative Areas (Grant Nos. JP24104004 and 18H05423) and a Grant-in-Aid for Scientific Research (A) (Grant No. 18H03673) from kaken. NO - The funders had no role in study design, data collection and analysis, decision to publish, or preparation of the manuscript.

**Competing interests:** The authors have declared that no competing interests exist.

regulation, and other functions [5]. In recent years, much effort has been devoted not only to understand the underlying mechanism of pattern formation by MTs in the cell but also to mimic those structures outside the cellular environment with a view to harness their emergent functions [6]. As a result, several strategies have been developed that have successfully produced diverse patterns of MTs such as asters, vortices, teardrops, rings, and bundles in *in vitro* conditions [7–13]. Previously we reported a helical alignment of MT structures by performing the polymerization of tubulins under temperature gradient with a spatial restriction [14,15]. The design of MT polymerization condition in an open system could help us to explore the diversity of the self-assembly behavior of MTs further. For instance, evaporation of droplet is a recurrent technique which allows different pattern formation of particles by evaporation of solvent in an open system. Convection flow (capillary and Marangoni flow) and different forces (collision, electrostatic interaction caused by the charges of the molecules, adhesion between the molecules and substrate) are responsible for the ultimate pattern formation by drying of the droplet studied so far for many synthetic as well as biomolecules [16–26]. Here, we report formation of spherulites from tubulins through polymerization into MTs by heating under water evaporation in a droplet like open system. MTs are found radially aligned in the millimeter-scale of patterns [27] confirmed from the observation under polarized optical and fluorescence microscopy. This work will facilitate not only to understand the self-assembly and resulting pattern formation of MTs but also new applications of cytoskeletal proteins in nanotechnology [28, 29].

## Materials and methods

### Sample preparation

**Tubulin purification.**   Tubulin was purified from porcine brains using a concentrated 1,4-piperazinediethanesulfonate (PIPES) (Sigma-Aldrich, USA) buffer according to the protocol established in previous work [30]. In brief, food grade porcine brains were purchased from a local slaughterhouse, conserved before use in ice-cold PBS (phosphate buffer saline) prepared by mixing 137 mM NaCl, 8.1 nM $Na_2HPO_4$, 2.68 mM KOH and 1.47 mM $KH_2PO_4$ (all reagents from WAKO Pure Chemical Corporation, Japan). High-concentration PIPES buffer and Brinkley BR buffer 1980 (BRB80) were prepared using the dipotassium salt of PIPES (Sigma-Aldrich, USA), and the pH was adjusted to 6.8 using KOH (WAKO Pure Chemical Corporation, Japan). The purity of tubulin was confirmed to be ~98–99% from the result of sodium dodecyl sulfate/polyacrylamide (WAKO Pure Chemical Corporation, Japan) electrophoresis [30].

**Preparation of labeled tubulin.**   ATTO 488-labeled tubulin was prepared using ATTO 488 succinimidyl ester (Sigma-Aldrich, USA) according to the standard protocol [31]. The labeling ratio of ATTO 488-modified tubulin was 1. This ratio was determined by measuring the absorbance of the tubulin and ATTO 488 at 280 nm and 501 nm, respectively using a UV spectrophotometer (Nanodrop 2000c, 2048-element linear silicon CCD array detector). Concentration and consequently ratio of tubulin and ATTO 488 was determined using Beer-Lambert law where the molar extinction coefficient,  of tubulin was 111500 L $cm^{-1}$ $mol^{-1}$ and ATTO-488 was 90000 L $cm^{-1}$ $mol^{-1}$. Path length was $l$ = 0.1 cm. BRB80 buffer was used as blank for the measurement.

**Formation of MT spherulites through heating and subsequent evaporation of tubulin solution.**   To obtain radially aligned MTs, tubulin dimers were polymerized in a droplet of solution with the evaporation of water. 2 μl of buffer solution (80 mM PIPES, 1 mM EGTA (DOJINDO MOLECULAR TECHNOLOGIES, INC. Japan), 1 mM $MgCl_2$ (WAKO Pure Chemical Corporation, Japan), pH~6.8) containing 1.0 mM GTP (WAKO Pure Chemical

Corporation, Japan) and tubulin dimer at a concentration of 180 μM was deposited on a cover glass of 0.17 mm thickness (24 mm x 60 mm, MATSUNAMI Inc.), which was mounted on a thermostatic plate set to 37 ˚C (MATS-55SF; Tokai Hit, Japan). The droplet was dried in the air by heating from bottom at 37 ˚C for ~40 min to evaporate the water and form spherulite patterns. For observation using a fluorescence microscope, the ATTO-488 labeled tubulin dimers were mixed with non-labeled dimers at a 1:4 molar ratio and used to prepare the corresponding ATTO-488 labeled MTs. The lowest ratio of labeled to unlabeled tubulin dimers (1:4 molar ratio) was used to provide signal intensity sufficient for observation of the MT structure formed by polymerization of tubulin dimers on the glass surface. To accomplish the experiment, the droplet of ATTO-488 labeled tubulin solution was evaporated and allowed to form spherulites. The droplet was then stabilized with taxol buffer (80 mM PIPES, 1 mM EGTA, 1 mM $MgCl_2$, 50 μM taxol (Sigma-Aldrich, USA), pH~6.8) and observed using fluorescence microscopy.

**Formation of reversible MT spherulites through subsequent heating and cooling.**   2 μl of 180 μM tubulin solution was subjected to evaporate at 37 ˚C for 10 min. After 10 min as the spherulite structures appeared, the cover glass containing the droplet was placed on ice at 0 ˚C immediately for 5 min without adding further buffer solution. Then as the structures were disappeared the droplet was subjected to evaporate again at 37 ˚C.

**Formation of the concentric ring-like pattern of MTs by modulation of water evaporation.**   2 μl of 180 μM tubulin solution was enclosed in PDMS sheet of 0.2 mm thickness (Wako Pure Chemical Corporation, Japan). For modulation of water evaporation, Shin-Etsu silicone oil (Shin-Etsu Chemical Co. Ltd; KF-968-100CS, Japan) was used to coat the droplet of tubulin solution.

## Measurement

**Polarized optical microscope measurement.**   Samples prepared according to the methods described above were observed by polarized optical microscope (BX51, Olympus, Japan) using a charge-coupled device (CCD) camera (Olympus) with 2×, 4× and 10× objective lens (Nikon). For the observation of optical retardation, a 530 nm sensitive color plate (U-TP530, Olympus) was placed between the sample and the analyzer.

**Fluorescence microscopy.**   The droplet of tubulin solution after drying and stabilizing with taxol buffer, was illuminated with a 100 W mercury lamp and visualized by an epifluorescence microscope (Eclipse Ti, Nikon) using 2×, 20× objective lens (Nikon). UV cut-off filter block (GFP-B: EX460-500, DM505, BA510-560; Nikon) was used in the optical path of the microscope. Images were captured using a cooled-CMOS camera (NEO sCMOS, Andor) connected to a PC. ND filter (ND32, 3.1% transmittance for GFP-B) was inserted into the illumination light path of the fluorescence microscope to reduce photobleaching of the samples.

**Small angle X-ray scattering (SAXS) measurements.**   The SAXS experiments were performed by using the Rigaku NANOPIX measurement system which has the highest level of small angle resolution ($Q_{min}$ to 0.02 $nm^{-1}$). The X-ray source was MicroMax-007 HF MR and the size was φ70 μm. CuKα radiation was used with 1.2 kW X-ray power for radiation. All the SAXS experiments were performed on the samples that were dried on polyimide film with 0.0075 mm thickness which was used as a sample holder. The image of the scattering pattern was obtained at a frame size of 775 x 385 pixels and a pixel size of 100 x 100 μm. The exposure time was 1 hour. CCD camera with semiconductor pixel sensor was used for the detection of diffraction patterns.

**Statistical procedures.**   The experiment was performed at least three times for each condition. The statistical analyses of diameter of spherulites were performed using Image J software

(ij152-win-java8, Center for Information Technology, Bethesda, Maryland, USA). The analyzed data of diameter of spherulites against time were plotted using OriginPro19 software (OriginPro version 2019, OriginLab Corporation, Northampton, MA, USA). The regression coefficient, $R^2$ values of slopes considered for the determination of growth rates at each temperature from the plots of diameter versus time were ~98–99%. The X-ray diffraction data were analyzed and plotted using Igor Pro 6.37 software (Wavemetrics, Inc. Lake Oswego, Oregon, USA).

## Results

To demonstrate the nucleation of tubulin dimers and subsequent MT growth in a droplet like spatial restriction under water evaporation, we designed an experimental system as illustrated in Fig 1A. 2 μl of tubulin solution at a concentration of 180 μM was deposited on a cover glass. The cover glass was then mounted on a thermostatic plate to maintain 37 ˚C by heating at the bottom surface of the glass. The top of the droplet was allowed to remain open at room temperature (25 ˚C). It is assumed that heating at 37 ˚C may cause the nucleation of tubulin dimers into MTs. The evaporation of water may induce alignment of the MTs resulting in radially aligned patterns, as shown in Fig 1A.

To detect the pattern formation, polymerization of tubulin (180 μM) was performed by heating tubulin solution at 37 ˚C according to the experiment as designed above. The radially aligned pattern formation was monitored by birefringence changes using polarized optical microscope. Maltese Cross patterns were observed under polarized optical microscope after 40 min of heating the solution (Fig 1B, 1C and 1D). The retardation of optical textures using a 530 nm sensitive color plate shown in Fig 1E suggests that MTs are aligned from the centers of the Maltese Crosses. A positive optical sign of birefringence, which generates a blue addition interference color when the long axis of the alignment of MTs is oriented parallel to the fast axis of the first order retardation plate [15, 32]. Above observations envisage us the formation of spherulite like structure from tubulin solution upon heating. To clearly visualize that the Maltase Cross pattern was formed from MTs alignment by tubulin polymerization, the droplet of tubulin solution was prepared using ATTO-488 labeled tubulin and the pattern formation was monitored using fluorescence microscope following the same procedure as described above after stabilizing with taxol buffer. The fluorescence microscopy images (Fig 1F and Fig 1G) of spherulites indicate the presence of aligned MTs that are nucleated from polymerization of tubulins. As taxol is only used for the stabilization of spherulite pattern to allow fluorescence microscope observation there is no possibility to form any taxol crystals like MTs bundle or aster as reported [33]. The fluorescence intensity coming from the dye labeled tubulins confirms the pattern formation by MTs from tubulin dimers. The length of MTs forming the pattern estimated after stabilizing with taxol buffer was 3.78 ± 1.14 μm (n = 100). The shorter length of MTs might be attributed to the dynamic instability of MTs resulted from change in different parameters (temperature and concentration of tubulin) suggesting a common effect involving tubulin-MT-water interaction [34].

Since the temperature gradient in the droplet in an open system may influence the radial alignment of MTs, the experiment was performed in a thermostatic chamber. In this chamber tubulin droplet on glass was placed on thermostatic plate and covered with another plate where temperature was maintained at 37 ˚C. This chamber keeps the air temperature constant throughout the space without any temperature gradient but still allows evaporation from the droplet. The growth of spherulites was monitored with time (S1 Fig). The formation of spherulites from MTs in thermostatic chamber indicates that temperature gradient is not necessary for radial alignment of MTs.

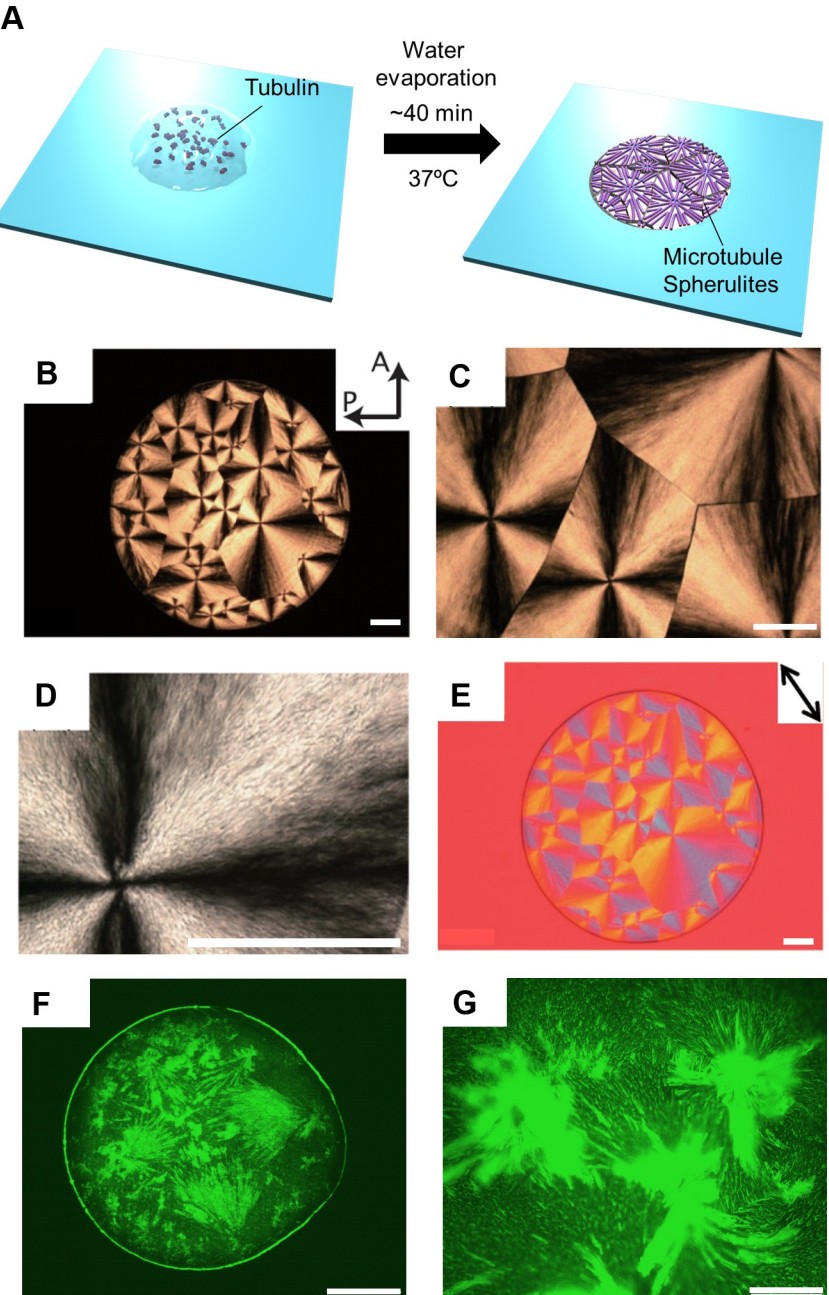

**Fig 1. Formation and observation of MT spherulites by polarized and fluorescence microscopy.** (A) Schematic illustration of the experimental setup for MTs spherulites formation from an evaporating droplet of the tubulin solution. Polarized optical microscopic images of MT spherulites under (B) 2×, (C) 4×, (D) 10× objective lens and (E) optical retardation. The volume of tubulin solution was used 2 μL for placing a droplet and concentration was 180 μM. The tubulin solution was evaporated at 37 °C. The images were captured after 40 min of evaporation. Scale bar: 1 mm. The arrows A and P represent the analyzer and polarizer directions. The arrow in (E) indicates the direction of the sensitive color plate. (F, G) Fluorescence microscopy images of MT spherulites stabilized with taxol buffer by evaporation at 37 °C. Scale bar: 1 mm (F) and 100 μm (G).

Fig 2A shows time-lapse polarized microscopy images of the emergence of the Maltese Crosses pattern in the droplet upon heating. After 10 min of heating at 37 ˚C, the pattern appeared (S2A Fig), and the diameter of the pattern increased with time at growth rate of

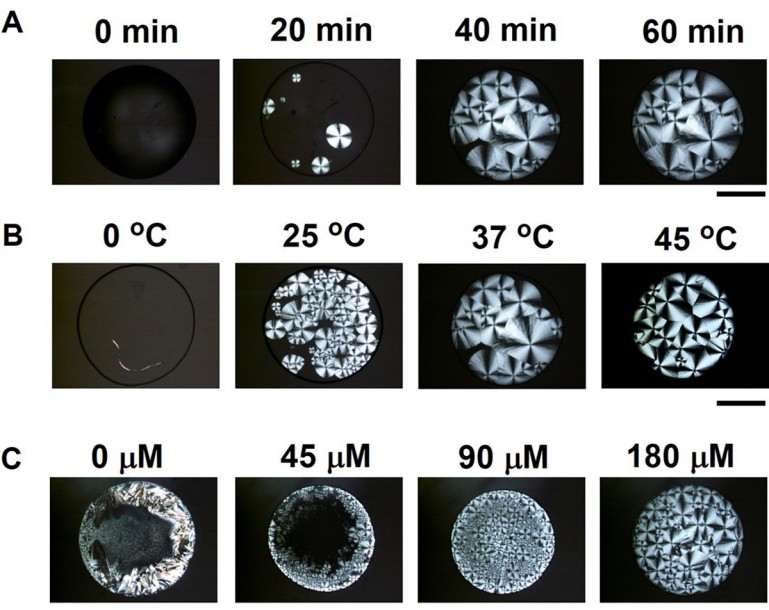

**Fig 2. Formation of MT spherulites depending on time, temperature and concentration of tubulin dimers.** (A) Time-lapse polarized optical microscopy images of MT spherulites. The tubulin solution was evaporated at 37 °C. (B) Growth of MT spherulites at different temperatures after 40 min of evaporation. The volume of tubulin solution was used 2 µL for placing the droplets and concentration was 180 µM. (C) Effect of concentration of tubulin solution (0 µM, 45 µM, 90 µM and 180 µM) on the formation of MT spherulites. The images were taken after 40 min of evaporation of the droplets at 37 °C. Scale bar: 1 mm.

~0.6 ± 0.03 µm s$^{-1}$ (average ± standard deviation (s.d.)) which is a little slower compared to the growth rate of MTs (~1.0 µm s$^{-1}$) evaluated using the equation $v(t) = k_{on}[T] - k_{off}$, where $k_{on}$ and $k_{off}$ are the association and dissociation rate constants at 37 °C respectively, and [T] is the concentration of free tubulin [14] (S2D and S2F Fig). The growth of spherulites was found to increase with increasing the temperature (Fig 2B) which indicates temperature is an important parameter to control the growth rate of Maltase Cross like pattern formation of MTs [34] (S2 Fig). However, the number of spherulites appeared shows opposite trend with growth rate at different temperatures (S2G Fig). Although the growth rate was lower at 25 °C (S2F Fig), the number of spherulites was found to increase sharply with time at 25 °C than that at higher temperatures (37 and 45 °C).

In this study we use concentration of tubulin solution (180 µM) for MT spherulite formation which is higher than that is required for treadmilling of MT [35]. To investigate the role of concentration of tubulin dimers on pattern formation, concentration of tubulin (0 µM, 45 µM, 90 µM and 180 µM) was varied as shown in Fig 2C. At 0 µM tubulin concentration, no Maltese Cross was observed except some crystal structures which grew from the drying of buffer solution containing GTP. The Maltese Crosses pattern formation in the droplets was observed at a certain tubulin concentration 45 µM which favored the velocity of growth of polymerization of tubulins into MTs [34, 36] and formed spherulite like structures at the edge of the droplet. The radially aligned structure formation at the edge of the droplet confirmed the replacement of the solute particles (tubulin dimes) by the capillary flow like convection flow similar to the so-common coffee ring effect at the elevated temperature [16–19] (Fig 2C). With further increasing the concentration of tubulin, nucleation points of spherulites appeared from the center of the droplet after the drying of the edge which indicates uniform deposition of tubulin dimers and consequently polymerized MTs. This finding indicates that MT

spherulite formation upon water evaporation follows the general rule of pattern formation in drying droplet which depends on the concentration of the tubulin dimer. The increasing tubulin dimer concentration may also alter the dynamics of solute-solvent interaction due to the different size, mass and charge of solute particles (salts, protein) that may prevent the crystal formation from buffer salts as observed in the case of 0 μM tubulin [18].

Alongside the driving force created by the convection flow in the solution, evaporation of water from the droplet plays an important role in the alignment of MTs into spherulites which were confirmed by sealing the droplet using silicone oil (Fig 3). The evaporation of water was thus regulated by oil coating and birefringence of MTs was observed under polarized optical microscope as a concentric ring-like pattern instead of spherulites. The optical sign of birefringence of this pattern was found opposite to the MT spherulite where a yellow subtraction interference color was observed parallel to the fast axis under the fast order retardation plate. The result indicates that evaporation is important for formation of spherulite pattern which changed to concentric ring upon modulation of evaporation.

To reversibly regulate the Maltese Crosses pattern in the spherulite like structure, the droplet of tubulin solution was cooled at 0 ˚C just after 10 min evaporation at 37 ˚C (Fig 4A). The patterns were disappeared within 5 min of cooling. The disappearance of the Maltese Crosses pattern might be attributed to the depolymerization of MTs into tubulin dimers at 0 ˚C [37]. The pattern appeared again upon heating the droplet at 37 ˚C (Fig 4A) indicating reversible spherulite formation of MTs.

To get detailed structural information on the alignment of MTs in the reversible Maltese Crosses pattern, small angle X-ray scattering (SAXS) measurements were performed. Two dimensional-scattering pattern and profile of spherulite appeared from evaporating droplet containing 180 μM tubulin solution were shown in Fig 4B. The 2D diffraction pattern of the radially aligned MTs after heating shows the tilt angle from its equator position indicating long-range ordering of MTs which is quite different from the isotropic orientation of MTs in solution (Fig 4B (i), (ii)). Sharp peaks arising from the near-meridional reflections with $d$ value of 2.23, 1.78, 1.58 and 1.12 nm (Fig 4C (ii); 1, 2, 3, 4 respectively) might be due to ordering of MTs in radially aligned pattern in evaporated state which differs from the wider peaks of MTs in solution [38] (Fig 4C (i); 1, 2, 3, 4 respectively). The higher-ordered alignments may give rise to these sharp peaks in the layer lines of detected $d$-spacing values which are similar to the previously reported values [39]. The $d$ values may also indicate the size of the tubulin monomers [40] which were closely packed resulting radial array of MTs. Upon cooling, the sharp peaks from the near-meridional reflections disappeared which confirms the spherulite was formed by the MTs (Fig 4C (iii)). Upon re-heating the droplet, the peaks at $d$~1.78 and 1.58 nm were regenerated (Fig 4C (iv): 1, 2). No reflection from 2D diffraction pattern and peaks were observed for buffer solution (S3 Fig) which confirms that the pattern is solely formed by MTs under water evaporation.

## Discussion

The mechanism of MT spherulite formation in the evaporating droplet is a very complicated one that cannot be elucidated by a single step. The evaporation of the droplet of tubulin solution indeed agrees with the so-called phenomenon coffee ring effect [16, 20, 22] by the convection flow and it may have some effect on the directional alignment of MTs. But the formation of radially aligned spherulite structure by the polymerized MTs still remains elusive. The possible explanation is that MT is considered as rigid rod with negative charges on its outer surface. In solution they stay apart from each other due to electrostatic repulsion. In droplet when evaporation occurs, the highly concentrated MTs polymerized from tubulin dimers come

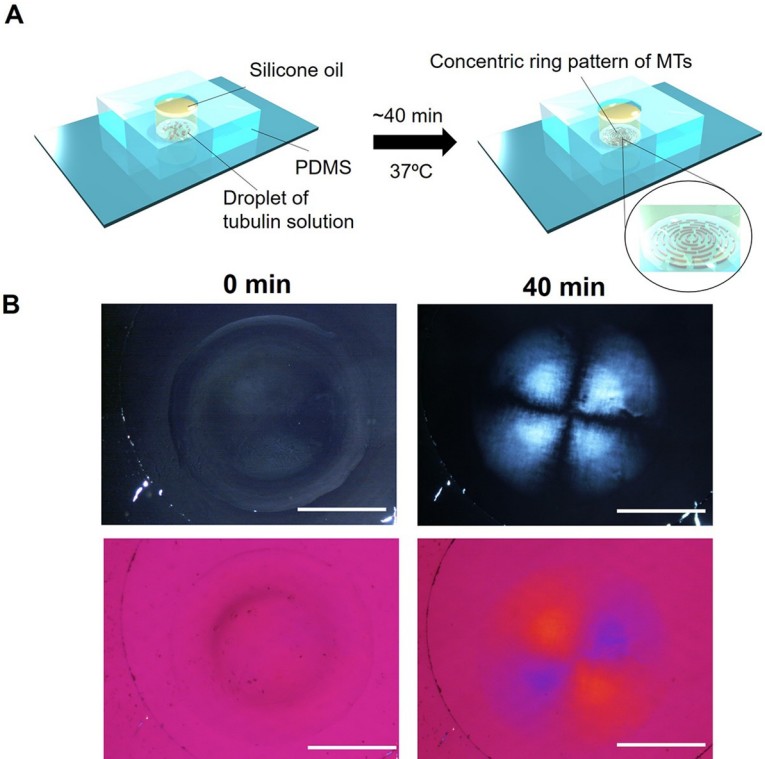

**Fig 3. Formation of concentric ring like pattern of MTs by modulating of evaporation of water.** (A) Schematic presentation of the experimental design of inhibition of water evaporation from a droplet of tubulin solution. The droplet was placed on a glass surface enclosed in polydimethylsiloxane (PDMS) sheet and sealed by silicone oil. (B) Polarized optical microscopy images of droplet of tubulin solution sealed with silicone oil as shown in the schematic presentation. At 0 min no birefringence and color change were observed under retardation plate. After 40 min of evaporation at 37 °C, birefringence was observed and color change under retardation plate indicates polymerization of MTs took place which formed a concentric ring like pattern. The volume of tubulin solution was used 2 µL for placing the droplet and concentration was 180 µM. Scale bar: 1 mm.

closer due to excluded volume effect and align themselves with each other. The high viscosity of the medium due to evaporation of water resulting in depletion force which favors the radial alignment of MTs in the crowded state [41, 42].

This type of radial alignment of MTs may give rise to the liquid crystalline assembly of MTs as observed in the previous studies of the dense solution of MTs [36]. The observed birefringence from spherulite structures at 37 °C indicates that tubulin dimers polymerized into MTs and aligned themselves showing birefringence can possess anisotropy. Further cold-induced depolymerization of MTs is accompanied by a loss of both anisotropy and birefringence due to the lack of alignment of MTs. Moreover, the 2D diffraction pattern of MT spherulite structure shows the anisotropic tilt angle in the equator position and sharp peaks in the near-meridional positions (Fig 4B (ii)) which are absent in case of the MT solution (Fig 4B (i)). These observations qualitatively depict that radial alignment of MTs may show liquid crystalline behavior.

Although, this is the very first reported data of MT spherulite structure from drying droplet of tubulin solution still there is a high scope to further characterize the structure to explore its functions. However, the results could suggest the significance of using the simple experimental technique of drying droplet solution to get the unique MT spherulite pattern compared to the previously observed MT polymorphism [43]. Accordingly, this technique could provide the

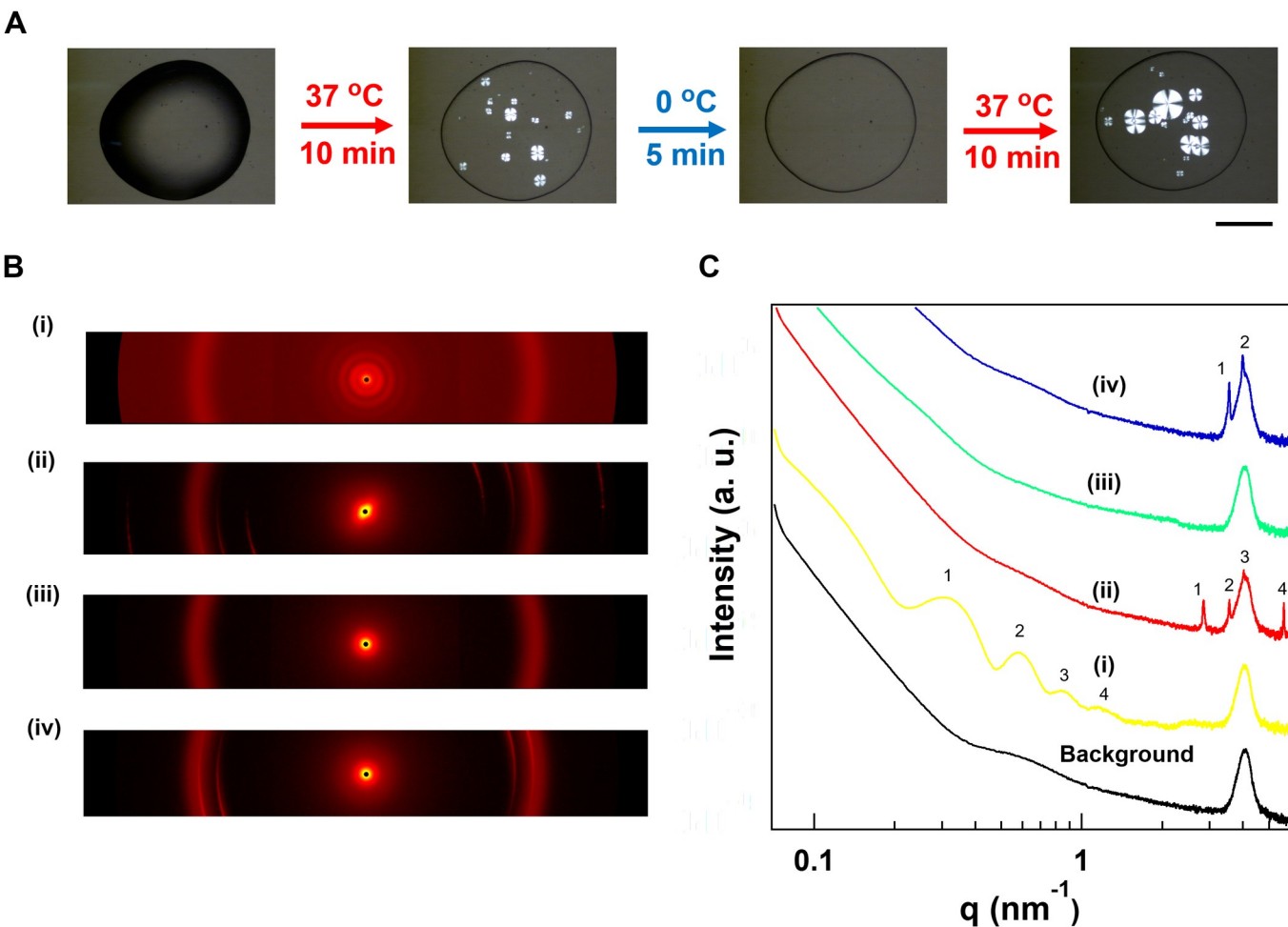

**Fig 4. Temperature dependent reversible MT spherulite formation and analysis of two-dimensional scattering pattern of them.** (A) Time-lapse polarized optical microscopy images of MT spherulites under reversible heating/cooling. Heating time 10 min and cooling time 5 min. Scale bar: 1 mm. (B) Two-dimensional scattering from droplets of tubulin solution at different conditions; (i) MT solution, (ii) spherulite after heating the droplet of tubulin solution at 37 °C for 40 min, (iii) after cooling the droplet of MT spherulite at 0 °C for ~5 min and (iv) after re-heating the droplet appearing MT spherulite again at 37 °C for 40 min. Profile of scattering curves of (i), (ii), (iii) and (iv) (C). The volume of tubulin solution was used 2 μL for placing the droplets and concentration was 180 μM in each case. Polyimide thin film was used as a sample holder in all experiments.

required information about the pattern formation of cytoskeletal proteins in a confined space and crowded state. The MT pattern formed in the specified conditions can provide clues to the tubulin polymorphs that may exist *in vivo* in the crowded state [44]. Extensive studies made previously indeed explained the pattern formation of MTs in dissipative conditions [8–11, 13, 14] but still, we can enhance our understanding to a greater extent about the dynamic instability and reaction rate of self-assembly of tubulin dimers in concentrated and droplet like confined state. The understanding could give us the knowledge on various functions of cytoskeletal proteins in the cell such as aster growth, cell division, cargo transport on MT network, etc [45]. Extrapolating the knowledge would help us to design bio-nano devices with various functionalities like in cell which could be utilized for many sophisticated applications such as artificial cell construction.

Even though there is always uncertainty to mimic the behavior of tubulin dimer *in vitro* to its behavior *in vivo* in the cytoplasmic milieu of the living cell. For that, we must overcome the limitations of the present study. The MT pattern formation from the evaporation of the

droplets depends on some external parameters such as the temperature of the substrate, the relative humidity of the evaporation environment and the air velocity as well as some internal parameters such as the size of the droplet, concentration and physicochemical properties of the solute particles in the solution. In this study, we successfully controlled some parameters such as temperature, the concentration of tubulin dimers but still, we have the scope to control the other parameters as mentioned above which is a future perspective. Moreover, for detail morphological characterization of the pattern, high-resolution techniques such as scanning electron microscope (SEM), transmission electron microscope (TEM) or solid-state nuclear magnetic resonance (NMR) study will be carried out which have successfully provided atomic-resolution insights into complex systems [46, 47]. Such study will help to explore valuable insights into the structural information and molecular level interaction in the MT spherulite pattern.

## Conclusions

In conclusion, by employing a simple experimental design, we demonstrate formation of spherulites from MTs by polymerizing tubulin dimers under water evaporation. MTs formed macroscopic Maltese Cross pattern in the spherulite structures where the growth rate of spherulites depends on temperature. A certain tubulin concentration is necessary for spherulite formation below which the spherulite with Maltese Cross pattern could not be observed. The spherulites emerge and disappear reversibly upon regulation of temperature. The alignment of MTs in the spherulite was confirmed from the results of fluorescence microscopy observations and also small angle X-ray scattering measurements. Previously, organized structure formation by cytoskeletal proteins filaments, in cooperation with motor proteins, has been studied both experimentally and theoretically [48, 49]. The spherulite structure of MTs presented here is much larger in size (millimeter) compared to the structures previously reported (micrometer scale ordering) [7]. Though self-assembly of MTs into different structure formation were studied [8–10, 13–15], MT spherulite formation by this simple approach leaves a future perspective to study its detailed mechanism and potential application. Thus, this work provides an easy route to novel molecular assembly and may help extend our current understanding of the self-assembly of cytoskeletal proteins under dissipative conditions. The presented spherulite structure may contribute to bio- and nanotechnological applications of cytoskeletal proteins [50].

## Supporting information

**S1 Fig. Formation of MT spherulite in a thermostatic chamber.** Time-lapse polarized optical microscopy images of MT spherulites by evaporating tubulin solution at 37 °C in a thermostatic chamber. The volume of tubulin solution was used 2 μL for placing the droplet and concentration was 180 μM. Scale bar: 1 mm.
(TIF)

**S2 Fig. Kinetics of MT spherulite formation depending on different temperatures.** (A) Time lapse polarized optical microscopy images of droplets of tubulin solution at different temperatures; 0 °C, 25 °C, 37 °C, 45 °C. Scale bar: 1 mm. The volume of tubulin solution was used 2 μL for placing the droplets and concentration was 180 μM. Growth of spherulite diameter with time at (B) 0 °C, (C) 25 °C, (D) 37 °C and (E) 45 °C. The different color indicates different spherulites and number of spherulites was considered five in each case. (F) Growth rate of spherulites at different temperatures. Error bar: s.d. The growth rate was determined from the slopes of the steeper region of the plots of diameter versus time. From the average of growth rate of five spherulites, the growth rate was estimated for each temperature. (G) And

number of spherulites with time at different temperatures.
(TIF)

**S3 Fig. Two-dimensional scattering salt crystals from dried droplet of buffer solution.** (A) Two-dimensional scattering from the droplet of buffer solution and (B) profile of scattering curve. The peak corresponds to the background coming from the sample holder. The volume of buffer solution (80 mM PIPES, 1 mM EGTA, 1 mM $MgCl_2$) was used 2 μL for placing the droplet.
(TIF)

## Acknowledgments

We thank Ms. Saori Mori, Dr. Hiromitsu Moriyama, Prof. Hiromu Saito, and Prof. Kiyotaka Shigehara (Tokyo University of Agriculture and Technology) for their kind assistance with the tubulin purification. We especially thank Mr. Riki Kato for his assistance in SAXS experiments (Fukuoka Institute of Technology).

## Author Contributions

**Conceptualization:** Jakia Jannat Keya, Daisuke Inoue, Tomomi Tani, Akira Kakugo, Kazuhiro Shikinaka.

**Formal analysis:** Jakia Jannat Keya, Nobuyoshi Miyamoto.

**Funding acquisition:** Akira Kakugo.

**Investigation:** Jakia Jannat Keya, Hiroki Kudoh, Nobuyoshi Miyamoto, Kazuhiro Shikinaka.

**Supervision:** Akira Kakugo.

**Writing – original draft:** Jakia Jannat Keya, Arif Md. Rashedul Kabir, Akira Kakugo, Kazuhiro Shikinaka.

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
