## [Decision Letter · Decision Letter 0]

18 Feb 2020

PONE-D-19-33923

Radial Alignment of Microtubules through Tubulin Polymerization in an Evaporating Droplet

PLOS ONE

Dear Prof. Kakugo,

Thank you for submitting your manuscript to PLOS ONE. After careful consideration, we feel that it has merit but does not fully meet PLOS ONE’s publication criteria as it currently stands. Therefore, we invite you to submit a revised version of the manuscript that addresses the points raised during the review process.

ACADEMIC EDITOR: Please insert comments here and delete this placeholder text when finished. Be sure to:

Indicate which changes are required versus recommended for acceptanceAddress any conflicts between the reviewsProvide specific feedback from your evaluation of the manuscript

We would appreciate receiving your revised manuscript by Apr 03 2020 11:59PM. To enhance the reproducibility of your results, we recommend that if applicable you deposit your laboratory protocols in protocols.io, where a protocol can be assigned its own identifier (DOI) such that it can be cited independently in the future. For instructions see: http://journals.plos.org/plosone/s/submission-guidelines#loc-laboratory-protocols

We look forward to receiving your revised manuscript.

Kind regards,

Ayyalusamy Ramamoorthy

Academic Editor

PLOS ONE

Journal Requirements:

1. Please remove your figures from within your manuscript file, leaving only the individual TIFF/EPS image files, uploaded separately.  These will be automatically included in the reviewers’ PDF.

Additional Editor Comments (if provided):

The authors should include limitations of the reported study and possible sources of errors

in the experimental measurements and sample preparations.

Discussion should be expanded to briefly mention the need for high-resolution

structural information on the reported materials. For example, solid-state

NMR can provide valuable insights into these materials. Such studies have successfully

provided atomic-resolution insights into complex systems like

bone materials (Mroue et al), cartilage (Xu et al) etc. These can be mentioned to

inspire the readers.

Reviewers' comments:

Reviewer's Responses to Questions

**Comments to the Author**

1. Is the manuscript technically sound, and do the data support the conclusions?

Reviewer #1: Partly

Reviewer #2: Yes

2. Has the statistical analysis been performed appropriately and rigorously? 

Reviewer #1: Yes

Reviewer #2: N/A

3. Have the authors made all data underlying the findings in their manuscript fully available?

Reviewer #1: Yes

Reviewer #2: Yes

4. Is the manuscript presented in an intelligible fashion and written in standard English?

Reviewer #1: Yes

Reviewer #2: Yes

5. Review Comments to the Author

Reviewer #1: COMMENTS TO AUTHOR:

The authors describe nucleation and polymerization of tubulin to form microtubules using concentration and temperature as the principal parameters. These parameters are also reported to control the pattern of microtubules in vitro. Overall, I consider this study is interesting and to be published. I recommend a minor revision for this manuscript prior to consideration for publication. A few minor comments are listed below.

1) In methods, authors mentioned “The droplet was dried in the air by heating from bottom for ~10 mins to evaporate the water and form spherulite patterns”. Provide the specific temperature (37 C or higher temperature) used for heating the sample.

2) Mention the rationale behind using the specific 1:4 molar ratio of labeled tubulin dimers to non-labeled dimers for fluorescence.

3) In line 107, authors mentioned “tubulin at a prescribed concentration”. Was the concentration same as used for reversible MT spherulite measurement i.e. 180 μM?

4) Figure 1A shows ~40 min for water evaporation which differs from what authors mentioned in the methods. Comment on this.

5) Figure-2, authors mentioned taxol itself can form pattern like MT, thus comment how the increasing concentration of tubulin is correlated to taxol concentration, and how the spherulites can be distinguished and quantified in figure-2. That said, are the observed pattern mentioned as crystal structures grew from buffer drying in Figure 2C (0 micromolar) are from taxol? If this is correct, comment why they do not see such structures in sample containing tubulins.

6) There exist a large body of literature on functional polymorphism of tubulins. The reported structurally distinct pattern generated in vitro do not provide any discussion in line with the function. Discussion referring to in vivo observations and how the MT polymorphism observed here and previously contribute to MT dynamics would be helpful. I recommend providing more discussion rather generalizing spherulite structure may contribute to bio- and nanotechnological applications of cytoskeletal proteins. A sub-section providing comparative discussion of spherulite and previously designed polymorphs of MT and their potential applications would benefit the scientific community.

Reviewer #2: The current manuscript describes the study involving the formation of spherulites from droplets of concentrated tubulin solution to microtubules. Authors discover that the evaporation induced formation of microtubules is radially aligned. Overall this study is well-executed experiments and well-written paper. I would recommend this manuscript for publication after minor revision

Did the authors performed any high-resolution microscopy to study the more detailed structure of the MTs, for example SEM or TEM.

These assemblies are very similar to typical liquid crystal assemblies. The authors should mention these points and explain whether these assemblies possess any liquid crystalline properties.

6. PLOS authors have the option to publish the peer review history of their article (what does this mean?). If published, this will include your full peer review and any attached files.

Reviewer #1: No

Reviewer #2: No

---

## [Author Response · Author response to Decision Letter 0]

14 Mar 2020

Academic Editor: 

Journal Requirements:

1. Please remove your figures from within your manuscript file, leaving only the individual TIFF/EPS image files, uploaded separately. These will be automatically included in the reviewers’ PDF.

Response: We now include author’s affiliations, headings, figure captions in the main manuscript and upload the figure files separately according to instructions provided by the Academic Editor in the decision email.

Additional Editor Comments (if provided):

The authors should include limitations of the reported study and possible sources of errors

in the experimental measurements and sample preparations.

Discussion should be expanded to briefly mention the need for high-resolution

structural information on the reported materials. For example, solid-state

NMR can provide valuable insights into these materials. Such studies have successfully

provided atomic-resolution insights into complex systems like

bone materials (Mroue et al), cartilage (Xu et al) etc. These can be mentioned to

inspire the readers.

Response: We thank editor for the valuable suggestions. We have added a brief discussion pointing out the issues in the main text in lines 361-376.

Reviewer 1:

1) In methods, authors mentioned “The droplet was dried in the air by heating from bottom for ~10 mins to evaporate the water and form spherulite patterns”. Provide the specific temperature (37 C or higher temperature) used for heating the sample.

Response: The specific temperature that is used for heating the sample is 37 oC. The information has been added in the main text in line 110.

2) Mention the rationale behind using the specific 1:4 molar ratio of labeled tubulin dimers to non-labeled dimers for fluorescence.

Response: We used a mixture of labeled and unlabeled tubulin dimers for observation under fluorescence microscopy. We used the lowest ratio of labeled to unlabeled tubulin dimers (1:4 molar ratio) that provides signal intensity sufficient for observation of the microtubule (MT) structure formed by polymerization of tubulin dimers on glass surface. The information has been added in the main text in line 113.

3) In line 107, authors mentioned “tubulin at a prescribed concentration”. Was the concentration same as used for reversible MT spherulite measurement i.e. 180 μM?

Response: The concentration was the same i.e.; 180 μM as used for reversible MT spherulite measurement. The concentration has been mentioned in the main text in line 107.

4) Figure 1A shows ~40 min for water evaporation which differs from what authors mentioned in the methods. Comment on this.

Response: We thank the reviewer for raising this point. In these experiments, the nucleation of tubulin polymerization and growth of spherulite structure was monitored from the beginning of heating the sample. It was observed that, with water evaporation, the nucleation of tubulin dimers into MTs and resulting spherulite structure started to form after ~10 min of heating the sample and continued to grow up to 40 min (Fig 4A and Fig S2A). Within 40 min, spherulite structures covered almost the droplet area resulting from water evaporation. To remove the discrepancy and keep similarity, we have omitted the time mentioned (10 min) and added the exact time information (40 min) in the main text in line 110.

5) Figure-2, authors mentioned taxol itself can form pattern like MT, thus comment how the increasing concentration of tubulin is correlated to taxol concentration, and how the spherulites can be distinguished and quantified in figure-2. That said, are the observed pattern mentioned as crystal structures grew from buffer drying in Figure 2C (0 micromolar) are from taxol? If this is correct, comment why they do not see such structures in sample containing tubulins.

Response: Your queries are quite reasonable, and we apologize for the incomplete information. Here we clarify the points to make the confusion clear.

For the experiment of MT spherulite formation under polarized light microscopy at 37 oC, we did not use taxol for any of the experiments. We only use taxol buffer to stabilize the spherulite structure after it was formed to allow observation of the pattern under fluorescence microscopy. Therefore, there is no possibility to form any crystal structure of taxol. To mention this point, we have modified the information in the main text in line 195. 

To clarify the result in Figure 2C, for 0 μM tubulin concentration, the droplet contains only BRB80 buffer (80 mM PIPES, 1 mM EGTA, 1 mM MgCl2) with 1 mM GTP which is the control for all experiments. The patterns are the aggregation of salt crystals resulting from water evaporation in the droplet. A brief explanation for the absence of such crystal structures in the presence of tubulin dimers has been also addressed in the main text in line 252 as follows 

“The increasing tubulin dimer concentration may also alter the dynamics of solute-solvent interaction due to the different size, mass and charge of solute particles (salts, protein) that may prevent the crystal formation from buffer salts as observed in the case of 0 μM tubulin (Carreón YJP, Ríos-Ramírez M, Moctezuma RE, González-Gutiérrez J. Sci Rep. 2018;8: 9580).”

6) There exist a large body of literature on functional polymorphism of tubulins. The reported structurally distinct pattern generated in vitro do not provide any discussion in line with the function. Discussion referring to in vivo observations and how the MT polymorphism observed here and previously contribute to MT dynamics would be helpful. I recommend providing more discussion rather generalizing spherulite structure may contribute to bio- and nanotechnological applications of cytoskeletal proteins. A sub-section providing comparative discussion of spherulite and previously designed polymorphs of MT and their potential applications would benefit the scientific community.

Response: We thank the reviewer for raising this specific point. Although, this is the first-time reported data of MT spherulite structure, still there is a high scope to further characterize the structure to quantify and explore its functions which we expect to do in the future study. We have added the following discussion providing a comparison on present and previous polymorphs of MTs and its potential applications in the main text in line 344 as follows:

“Although, this is the very first reported data of MT spherulite structure from drying droplet of tubulin solution still there is a high scope to further characterize the structure to explore its functions. However, the results could suggest the significance of using the simple experimental technique of drying droplet solution to get the unique MT spherulite pattern compared to the previously observed MT polymorphism (Burton PR. BT-Cell and Muscle Motility. In: Dowben RM, Shay JW, editors. Boston, MA: Springer US; 1981. pp. 289–333). Accordingly, this technique could provide the required information about the pattern formation of cytoskeletal proteins in a confined space and crowded state. The MT pattern formed in the specified conditions can provide clues to the tubulin polymorphs that may exist in vivo in the crowded state (Ellis RJ. Trends Biochem Sci. 2001;26: 597–604). Extensive studies made previously indeed explained the pattern formation of MTs in dissipative conditions but still, we can enhance our understanding to a greater extent about the dynamic stability and reaction rate of self-assembly of tubulin dimers in concentrated and droplet like confined state. The understanding could give us the knowledge on various functions of cytoskeletal proteins in the cell such as aster growth, cell division, cargo transport on MT network, etc (Mitchison T, Wühr M, Nguyen P, Ishihara K, Groen A, Field CM. Cytoskeleton. 2012;69: 738–750). Extrapolating the knowledge would help us to design bio-nano devices with various functionalities like in cell which could be utilized for many sophisticated applications such as artificial cell construction.” 

Reviewer :2

Did the authors performed any high-resolution microscopy to study the more detailed structure of the MTs, for example SEM or TEM.

Response: We appreciate these comments and recommendations. Here, we report for the first-time the MT spherulite structure formation using the drying droplet phenomenon which is very sensitive to the relative humidity of the experimental environment, air velocity and some other internal parameters. Unfortunately, such sensitivity has limited our observed results for the detailed study using high-resolution techniques. We hope we will be able to control all the parameters to reveal the spherulite pattern formation of MTs and study the molecular level interaction using high-resolution techniques such as SEM, TEM or NMR. The discussion has been added as a future perspective in the main text in line 370.

These assemblies are very similar to typical liquid crystal assemblies. The authors should mention these points and explain whether these assemblies possess any liquid crystalline properties.

Response: To address this point, we have added an explanation in the main text in the discussion section in line 334 as follows

“This type of radial alignment of MTs may give rise to the liquid crystalline assembly of MTs as observed in the previous studies of the dense solution of MTs (Hitt AL, Cross AR, Williams RC. J Biol Chem. 1990;265: 1639–1647). The observed birefringence from spherulite structures at 37 oC indicates that tubulin dimers polymerized into MTs and aligned themselves showing birefringence can possess anisotropy. Further cold-induced depolymerization of MTs is accompanied by a loss of both anisotropy and birefringence due to the lack of alignment of MTs. Moreover, the 2D diffraction pattern of MT spherulite structure shows the anisotropic tilt angle in the equator position and sharp peaks in the near-meridional positions (Fig 4B (ii)) which are absent in case of the MT solution (Fig 4B (i)). These observations qualitatively depict that radial alignment of MTs may show liquid crystalline behavior.”

---

## [Editor Report · Decision Letter 1]

23 Mar 2020

Radial Alignment of Microtubules through Tubulin Polymerization in an Evaporating Droplet

PONE-D-19-33923R1

Dear Dr. Kakugo,

We are pleased to inform you that your manuscript has been judged scientifically suitable for publication and will be formally accepted for publication once it complies with all outstanding technical requirements.

With kind regards,

Ayyalusamy Ramamoorthy

Academic Editor

PLOS ONE

Additional Editor Comments (optional):

Revisions are satisfactory.

The authors have addressed the major concerns and the revised manuscript can be published.
---

## [Editor Report · Acceptance letter]

26 Mar 2020

PONE-D-19-33923R1 

Radial Alignment of Microtubules through Tubulin Polymerization in an Evaporating Droplet 

Dear Dr. Kakugo:

I am pleased to inform you that your manuscript has been deemed suitable for publication in PLOS ONE. Congratulations! Your manuscript is now with our production department. 

With kind regards,

on behalf of

Dr. Ayyalusamy Ramamoorthy 

Academic Editor

PLOS ONE